# Chemokine CXCL10 Modulates the Tumor Microenvironment of Fibrosis-Associated Hepatocellular Carcinoma

**DOI:** 10.3390/ijms23158112

**Published:** 2022-07-23

**Authors:** Elisa F. Brandt, Maike Baues, Theresa H. Wirtz, Jan-Niklas May, Petra Fischer, Anika Beckers, Björn-Carsten Schüre, Hacer Sahin, Christian Trautwein, Twan Lammers, Marie-Luise Berres

**Affiliations:** 1Department of Internal Medicine III, RWTH Aachen University, 52074 Aachen, Germany; elisa.brandt@rwth-aachen.de (E.F.B.); thwirtz@ukaachen.de (T.H.W.); pfischer@ukaachen.de (P.F.); anibeckers@ukaachen.de (A.B.); hacersahin@gmx.de (H.S.); ctrautwein@ukaachen.de (C.T.); 2Institute for Experimental Molecular Imaging, University Hospital RWTH Aachen, 52074 Aachen, Germany; maike.baues@rwth-aachen.de (M.B.); jmay@ukaachen.de (J.-N.M.); carsten.schuere@rwth-aachen.de (B.-C.S.); tlammers@ukaachen.de (T.L.); 3Center for Integrated Oncology Aachen Bonn Cologne Duesseldorf (CIO ABCD), 52074 Aachen, Germany

**Keywords:** CXCL10, hepatocellular carcinoma, tumor microenvironment, tumor-associated immune response, tumor cell proliferation, angiogenesis, chemokine/chemokine receptor network

## Abstract

Hepatocellular carcinoma (HCC) constitutes a devastating health burden. Recently, tumor microenvironment-directed interventions have profoundly changed the landscape of HCC therapy. In the present study, the function of the chemokine CXCL10 during fibrosis-associated hepatocarcinogenesis was analyzed with specific focus on its impact in shaping the tumor microenvironment. C57BL/6J wild type (WT) and *Cxcl10* knockout mice (*Cxcl10*^−/−^) were treated with diethylnitrosamine (DEN) and tetrachloromethane (CCl_4_) to induce fibrosis-associated HCCs. *Cxcl10* deficiency attenuated hepatocarcinogenesis by decreasing tumor cell proliferation as well as tumor vascularization and modulated tumor-associated extracellular matrix composition. Furthermore, the genetic inactivation of *Cxcl10* mediated an alteration of the tumor-associated immune response and modified chemokine/chemokine receptor networks. The DEN/CCl_4_-treated *Cxcl10*^−/−^ mice presented with a pro-inflammatory tumor microenvironment and an accumulation of anti-tumoral immune cells in the tissue. The most striking alteration in the *Cxcl10*^−/−^ tumor immune microenvironment was a vast accumulation of anti-tumoral T cells in the invasive tumor margin. In summary, our results demonstrate that CXCL10 exerts a non-redundant impact on several hallmarks of the tumor microenvironment and especially modulates the infiltration of anti-tumorigenic immune cells in HCC. In the era of microenvironment-targeted HCC therapies, interfering with CXCL10 defines a novel asset for further improvement of therapeutic strategies.

## 1. Introduction

Hepatocellular carcinoma (HCC) is one of the most common and lethal malignant tumors, accounting for ~90% of liver cancer cases. The main risk factor for HCCs is chronic liver disease. During the early promotion and progression of HCC, symptoms are sparse, often masked by the general conditions associated with the underlying chronic liver disorder, thus hindering early cancer detection. Diagnosis of HCC at advanced cancer stages therefore contributes to its high mortality rate. Currently, treatment options in advanced cancer stages are limited [1,2]. Clearly, new therapeutic strategies are required to reduce the incidence and progression of HCC and to improve patients’ survival, while also considering the impact of such approaches on the underlying chronic liver disease. In this setting, the recent triumph of combinational therapy with immune checkpoint inhibitors (ICI) and anti-angiogenesis agents (i.e., atezolizumab plus bevacizumab) for advanced HCC treatment highlights the pivotal role of the tumor microenvironment (TME) [3]. Unfortunately, still only 27.3% of patients exhibit an objective response (partial response or complete response) to this treatment. Hence, the overall long-term prognosis of patients with HCC still remains poor.

Chemokines are abundantly present in the TME and play key roles for cancer hallmarks such as tumor cell proliferation, angiogenesis, tumor stroma composition, and the tumor-associated immune response. The chemokine receptor CXCR3 and its interferon (IFN)-gamma inducible ligands CXCL11, CXCL10 and CXCL9 have distinct biological roles in the TME [4]. It has been acknowledged so far that the CXCR3/ligand signaling axis modulates pro- or anti-tumorigenic features of tumor and endothelial cells. These beneficial as well as harmful effects are tied to the primary underlying chronic disorder and the tumor entities. For instance, CXCR3 has a pro-tumorigenic effect in prostate cancer [5], pancreatic cancer [6] and gliomas [7,8]. CXCR3-expressing tumor-associated endothelial cells exert anti-tumorigenic effects [9] and have been described as anti-proliferative and pro-apoptotic in renal cancer [10], breast cancer [11] and myeloma [12]. Furthermore, the CXCR3–CXCL10 axis orchestrates the recruitment and polarization of several immune cell subtypes in the TME and thereby shapes the inflammatory tumor milieu [4]. The effects of CXCL10 on tumorigenesis and TME hence appear to be dependent on the pathological entity, the affected cell type and the quantity of CXCL10 secretion by various disease-specific immune cell subpopulations as well as structural features of the chemokine (e.g., non-truncated or truncated CXCL10 [13]).

In the liver, the CXCR3-ligand axis targets various intrahepatic cell types (e.g., hepatocytes, LSECs, hepatic stellate cells), liver resident and infiltrating immune cells as well as tumor cells [4]. Specifically, CXCL10 has been shown by our group and others to promote liver fibrosis in men and mice [14,15]. However, less is known on its functional role in end-stage complications of fibrosis, e.g., hepatocarcinogenesis. In the present study, we show that CXCL10 has an impact on several hallmarks of the HCC microenvironment and reveal a pro-tumorigenic function of CXCL10 in a murine fibrosis-associated HCC model. Hence, the modulation of the TME via CXCL10 might represent a novel asset for future therapy strategies.

## 2. Results

### 2.1. CXCL10 Is Critical for Fibrosis-Associated Hepatocarcinogenesis

Intrahepatic *CXCL10* mRNA expression in human HCC patients was analyzed using the gene expression profiling interactive analysis (GEPIA) database [16]. HCC patients express significantly higher levels of *CXCL10* in comparison to healthy controls (Figure 1A). Yet, there seems to be no significant differences between HCC tissue sample stratified by etiology of chronic liver disease in a second cohort of patients using the cBioPortal database (Appendix A). To assess the functional relevance of CXCL10 during hepatocarcinogenesis, mice proficient for Cxcl10 (C57Bl/6J WT) or with genetic deletion of *Cxcl10* (C57Bl/6J *Cxcl10*^−/−^) were analyzed. Here, we used an established model of fibrosis-associated hepatocarcinogenesis with a combination of diethylnitrosamine (DEN) and a low dosage of the hepatotoxin carbon tetrachloride (CCl_4_) recapitulating hallmarks of human HCC [17]. The untreated mice served as healthy control, the CCl_4_-treated mice served as fibrotic control (Figure 1B). In accordance with human HCC tissue, tumorigenic and surrounding non-tumorigenic liver tissue of the DEN/CCl_4_-treated WT mice showed an enhanced expression of *Cxcl10* compared to liver tissue of untreated mice. Of note, low dose CCl_4_ treatment alone did not result in a significant induction of *Cxcl10* in the absence of tumorous tissue (Figure 1C). To localize regions of higher Cxcl10 protein expression within the tissue, Cxcl10 immunohistological staining of liver sections of untreated and tumor-bearing wildtype mice were performed. As proof of principle, *Cxcl10*-deficient mice showed no specific staining of Cxcl10 in any conditions (Figure 1D). Tumor-bearing livers of WT mice compared to the untreated wild-type setting displayed a strong Cxcl10 accumulation especially in cancer cells in tumor regions as well as in hepatocytes in non-tumorous tissue in close proximity to the portal fields and fibrotic septa and in hot spots of immune cell infiltration (Figure 1D). Next, we set out to assess the functional impact of Cxcl10 expression on tumor burden in mice by subjecting mice with or without genetic deletion of *Cxcl10* to our tumor model. Compared to WT counterparts, *Cxcl10*^−/−^ mice showed a decreased tumor burden as evidenced by reduced total number and size of tumors (Figure 1E,F). In both genotypes, tumors displayed the typical histopathological feature of low-differentiated HCC with pleomorphic tumor cells and a typical shift of cytoplasm/cell nucleus ratio (Figure 1G). Furthermore, qRT-PCR analysis revealed a significantly elevated *Afp* expression as a marker of tumor cell de-differentiation in WT and *Cxcl10*^−/−^ HCC tissue in comparison to the surrounding tissue (Figure 1H).

### 2.2. Cxcl10 Deficiency Is Associated with a Reduced Tumor Cell Proliferation and a Remodeling of the Tumor-Associated Extracellular Matrix Composition

The CXCR3-ligand axis has been implicated in tumor cell proliferation of various cancer types [4]. HCC tissue of the DEN/CCl_4_-treated *Cxcl10*^−/−^ mice showed a slight reduction of proliferative Ki67+ tumor cells in comparison to tumorigenic WT tissue (Figure 2A). To test whether CXCL10 might directly influence tumor cell proliferation as tumor cells can express CXCR3, we performed a proliferation assay with the human hepatoma cell line HUH7. HUH7 cells were stimulated with CXCL10 and CXCL9. Additionally, cells were treated with a CXCR3 inhibitor. Stimulation with CXCL9 or CXCL10 fostered HUH7 proliferation (Figure 2B). Interestingly, pharmacological blockade of CXCR3 reversed this effect. Accordingly, the data suggest that the slight reduction of tumor cell proliferation in *Cxcl10*^−/−^ mice is a direct effect of the impairment of CXCR3–CXCL10 signaling. In vivo, however, one might speculate that the pro-proliferative effects of the functionally intact CXCR3–CXCL9 axis may compensate for the effect of CXCL10 deficiency in this aspect (Figure 2B), although there is no compensatory increase in *CXCL9* mRNA expression in *Cxcl10*^−/−^ mice in either condition (Appendix A). Nevertheless, taken together, this slight overall impact on tumor cell proliferation might not fully explain the significant reduction of tumor burden in *Cxcl10*^−/−^ mice.

Progression of HCC is not solely determined by biological features of cancer cells, such as their proliferation rate, but is also restricted and modulated by a TME composed of a network of immune and stroma cells. Of note, the chemokine CXCL10 has already been implicated in acute and chronic fibrosis as well as carcinogenesis [4,18,19,20]. Next, we set out to characterize the relevance of CXCL10 for the fibrotic non-tumorigenic microenvironment and the tumor stroma composition during hepatic carcinogenesis using HCC- bearing WT and *Cxcl10*^−/−^ mice. The CCl_4_ or DEN/CCl_4_-treated WT and *Cxcl10*^−/−^ mice display a similar fibrotic phenotype in non-tumorigenic liver tissue as evidenced via Sirius red staining (Figure 2C). To gain a deeper insight into the tumor stroma distribution and intra-tumoral extracellular matrix (ECM) composition, liver sections of the DEN/CCl_4_-treated WT and *Cxcl10*^−/−^ mice were stained for collagen IV (COLIV) and αSMA. Calculation of the area fraction demonstrated an overall upregulated level of αSMA as well as COLIV in tumor and tumor-surrounding tissue of the DEN/CCl_4_-treated *Cxcl10*^−/−^ mice compared to their WT counterparts (Figure 2D). The difference is particularly evident in the tumor stroma as αSMA forms a distinct tumor capsule. To analyze the TME ECM composition in more detail, we further measured the expression level of collagen 1α1 via qRT-PCR in tumor and tumor-surrounding tissue of the DEN/CCl_4_-treated WT and the *Cxcl10*^−/−^ mice. Unexpectedly, the DEN/CCl_4_-treated WT and *Cxcl10*^−/−^ mice displayed equivalent collagen 1α1 levels (Figure 2E).

### 2.3. Deletion of Cxcl10 during Hepatocarcinogenesis Attenuates Tumor-Associated Neovascularization and the Survival of Small Mononuclear Infiltrating Cells

Since tumor-associated angiogenesis is a hallmark of cancer progression, we analyzed the tumor vascularization of tumorigenic WT and *Cxcl10*^−/−^ mice [18]. With regard to tumor vascularization, the DEN/CCl_4_-treated *Cxcl10*^−/−^ mice displayed a reduced tumor-associated angiogenesis compared to WT counterparts (Figure 3A). Next, we evaluated whether this association between the genetic deletion of *Cxcl10* and tumor-associated angiogenesis is a statistical phenomenon or whether a functional relationship exists between these parameters. Hence, to get a deeper insight in the *Cxcl10*^−/−^ dependent tumor-associated angiogenesis, we performed another proliferation assay with the human endothelial cell line HUVEC. Consistent with our in vivo results, the stimulation of HUVEC cells with CXCL9 and CXCL10 enhanced their proliferation rate. The inhibition of CXCR3 reverted this effect (Figure 3B). This finding indicates that the reduced tumor-associated angiogenesis in the *Cxcl10*^−/−^ mice is linked to direct effects of an impaired CXCL10–CXCR3 signaling on endothelial cells.

In addition to the tumor-associated angiogenesis, cell death is another hallmark of cancer progression. Hence, we next set out to evaluate the amount of cell death in tumor tissue and the tumor microenvironment via a TUNEL assay and immunohistological staining of activated cleaved caspase 3. Cleaved caspase 3 staining demonstrated that deletion of *Cxcl10* leads to a minor enhancement of tumor cell death in the DEN/CCl_4_-treated *Cxcl10*^−/−^ mice in comparison to the DEN/CCl_4_-treated WT mice (Figure 3C). Interestingly, the DEN/CCl_4_ treated *Cxcl10*^−/−^ mice displayed a reduced number of TUNEL-positive and cleaved caspase 3-positive small mononuclear infiltrating cells in the area of the portal field in liver tissue in comparison to the DEN/CCl_4_-treated WT mice (Figure 3D).

Altogether, these data suggest that CXCL10 shapes the TME affecting tumor cell proliferation, angiogenesis and death of infiltrating stromal immune cells.

### 2.4. CXCL10 Modulates the Tumor-Associated Immune Cell Infiltration

As observed in other tumor entities, CXCL10 modulates leukocyte migration to the tumor site thereby governing anti- or pro-tumorigenic activity [4]. To test whether the genetic deletion of *Cxcl10* has an impact on the tumor-associated immune response during fibrosis-associated hepatocarcinogenesis, we evaluated potential differences in the immune cell repertoire in the liver tissues of the WT and *Cxcl10*^−/−^ mice via multiparametric flow cytometry and immunohistological staining. Interestingly, we observed a tumor-independent enhancement of CD45+ cells in the CCl_4_ and DEN/CCl_4_-treated *Cxcl10*^−/−^ mice compared to the WT counterparts (Figure 4A). Detailed analysis of different immune cell subtypes of the DEN/CCl_4_-treated WT and *Cxcl10*^−/−^ mice revealed an increased amount of NK cells, NKT cells, T cells (CD4+, CD4+ Treg and cytotoxic CD8+ T cells), neutrophils, plasmacytoid dendritic cells (pDCs) and myeloid cells (Figure 4B). Histological characterization of the spatial distribution of CD3+ T cells showed a defined accumulation of T cells in the surrounding tissue of HCCs in both genotypes (Figure 4C). Tumorigenic *Cxcl10*^−/−^ mice, however, displayed a strikingly increased number of T-cell hot spots in the surrounding tissue of HCCs compared to their WT counterparts.

Next, we used a multiplex staining setup to comprehensively study the spatial distribution of T cells and their subsets in liver tissue of the DEN/CCl_4_-treated WT and *Cxcl10*^−/−^ mice. Liver sections were stained for CD4+, CD8+ and Ki67+ cells, as well as collagen IV and αSMA (Figure 5B). Whole slide scans were acquired to identify tumor lesions as the regions of interest. To verify the localization of the tumor lesions, we used H and E staining of the same slides (Figure 5A). Utilizing the inForm analysis program, an algorithm was trained, enabling for tissue segmentation (tumor tissue, adjacent tissue, invasive margin and blood vessel), cell segmentation, cell phenotyping and cell positivity scoring (Appendix A). While phenotyping is a “learn-by-example” method, scoring is based on a set threshold for the fluorescence intensity, which has to be passed, for a cell to be assigned positive. The scoring analysis also allows a differentiation between the different tissue segments. Consistent with the conventional CD3 staining, multiplex analysis unraveled a strong enhancement of CD4+ and CD8+ T cells specifically in the invasive margin in tumorigenic liver tissue of *Cxcl10*^−/−^ mice (Figure 5C).

Taken together, these data suggest that the enhancement of tumor-fighting immune cells in the DEN/CCl_4_-treated *Cxcl10*^−/−^ mice shapes the tumor microenvironment in a tumor-diminishing manner, which fostering immune surveillance and tumor rejection.

### 2.5. Cxcl10^−/−^ Immune Cells Modulates the Tumor Microenvironment in an Anti-Tumoral Manner

Since tumorigenic *Cxcl10*^−/−^ mice display a strong enhancement of anti-tumoral immune cells, we proceeded to profile whether these cells tip the balance of the tumor microenvironment to an overall anti-tumoral immune response. First, we comprehensively characterized the cytokine pattern. In line with the previous results, qRT-PCR revealed an elevated expression of *Il6*, *Infy*, *Il1ß* and *Il12* in the tumor tissue of the DEN/CCl_4_-treated *Cxcl10*^−/−^ mice in comparison to the DEN/CCl_4_-treated WT mice (Figure 6A). Furthermore, tumor tissue of the *Cxcl10*^−/−^ mice displayed a strong enhancement of cytotoxic cytokines (*perforin* and *granzym B*) in comparison to their WT counterparts, which is probably related to the increased number of cytotoxic CD8+ T cells and NK cells (Figure 6B). Next, we assessed the T-cell exhaustion/dysfunction and immunosuppressive status of T cells via *Lag3* and *Tim3* and *Pd-1/Pd-l1* and *Ctla4*. Interestingly, the DEN/CCl_4_-treated WT mice showed a significant enhanced expression of *Lag3* in the surrounding tissue in comparison to tumor tissue. Liver tissue of the DEN/CCl_4_-treated *Cxcl10*^−/−^ mice displayed the same level of *Lag3* in the surrounding and tumor tissue. The expression level of *Tim3* was unaltered in the tumorigenic WT and *Cxcl10*^−/−^ mice (Figure 6C). Immune checkpoint molecules (*Pd-l1* and *Ctla4*, Figure 6D) showed the same level in the DEN/CCl_4_-treated *Cxcl10*^−/−^ mice and WT mice. Moreover, immunofluorescence staining of PD-1 revealed the same distribution of PD-1+ immune cells in the tumor stroma WT and *Cxcl10*^−/−^ mice.

Taken together, these data suggest that the enhancement of anti-tumoral immune cells in the DEN/CCl_4_-treated *Cxcl10*^−/−^ mice shapes the tumor microenvironment in a tumor-suppressive manner.

### 2.6. Genetic Deletion of Cxcl10 Modulates the Chemokine Receptor/Chemokine Network in the Tumor Microenvironment

The chemokine receptor/chemokine network coordinates immune cell trafficking and has a pivotal role for the migration pattern of immune cells to the tumor side [22].

Recently, dipeptidyl peptidase (DPP) IV has been reported to attenuate anti-cancer immunity via chemokine cleavage [23]. DPPIV is a peptidase that, among other things, can post-translationally modify CXCL10 via cleavage, resulting in a competitive antagonist for CXCR3-mediated signaling. In addition to CXCL10, DPPIV modulates a big batch of additional cytokines [24]. In our setting, however, the deletion of *Cxcl10* during fibrosis-associated hepatocarcinogenesis did not alter the distribution or amount of the intrahepatic DPPIV expression nor the serum level of DPPIV (Figure 7A,B), arguing against a specific role for DPPIV-dependent cytokine modulation in our model.

To further dissect the role of CXCL10 within the chemokine/chemokine receptor network during hepatocarcinogenesis, we analyzed some of the most important chemokines and chemokine receptors of the DEN/CCl_4_-treated WT and *Cxcl10*^−/−^ mice. It is striking that the DEN/CCl_4_-treated WT mice synthesize higher levels of chemokines on average in comparison to the DEN/CCl_4_-treated *Cxcl10*^−/−^ mice. Interestingly, comparison between both tissue types in the DEN/CCl_4_-treated WT and *Cxcl10*^−/−^ mice revealed a reduced chemokines accumulation in WT tumor tissue compared to WT surrounding tissue. The DEN/CCl_4_-treated *Cxcl10*^−/−^ mice, in turn, displayed a significant enhancement of chemokines in tumor tissue in comparison to the surrounding tissue (*Ccl5*, *Ccl2* and *Ccl7*) (Figure 7B). This chemokine gradient may therefore explain the increased accumulation of immune cells in tumorigenic liver tissue, since a tumor-specific attraction takes place. In addition, the tumor tissue of the DEN/CCl_4_-treated WT mice shows a reduction of chemokine receptors levels compared to the surrounding tissue of the same mouse (*Ccr5*, *Ccr2*, *Ccr7*, *Ccr1* and *Cxcr4*) (Figure 7D). This is probably related to the reduction of immune cells in the tumor tissue of the DEN/CCl_4_-treated WT mice in comparison to the surrounding tissue of these mice. In summary, deletion of *Cxcl10* in tumor-bearing mice leads to an overall reduction of chemokines but to a specific accumulation at the tumor site.

Consequently, the deletion of *Cxcl10* not only changes the immune cell repertoire and the tumor-associated inflammatory tumor microenvironment, but also impacts the overall chemokine/chemokine receptor network during hepatic carcinogenesis.

## 3. Discussion

Cancer immunotherapies have revolutionized the field of oncology. Unfortunately, individual responses are highly variable, and markers to predict response rates in patients are lacking. This disparity might be supported by a highly variable and dynamic immune tumor microenvironment [19]. Due to the multifaceted role that chemokine/chemokine receptor interaction plays in cancer promotion and progression, the chemokine system has been widely recognized as an important target for cancer immunotherapy. During acute and chronic liver injury as well as carcinogenesis, several CXCL10-associated cells shape the intrahepatic microenvironment in men and mice. Several studies have shown that CXCR3 and its ligands modulate the tumor microenvironment in a pro- or anti-tumorigenic manner. These beneficial as well as harmful effects are tied to the distinct main effector cell type, the pre-existing disease and the cancer type [4,20]. In general, the intrahepatic tumor progression is linked to the dynamic process of neovascularization, chronic liver inflammation and the crosstalk between tumor cells, peritumoral parenchymal cells as well as non-parenchymal cells, e.g., infiltrating immune cells [25].

In the present study, the function of CXCL10 during fibrosis-associated hepatocarcinogenesis was investigated. The initial findings were a reduced hepatocarcinogenesis as well as slightly reduced tumor cell proliferation and moderately enhanced tumor cell death in the DEN/CCl4-treated *Cxcl10*^−/−^ mice compared to identically treated WT mice. Furthermore, the DEN/CCl4-treated *Cxcl10*^−/−^ mice showed a reduced tumor-associated angiogenesis and an altered EMC/tumor stroma composition. In the literature, it is described that the CXCR3/CXCL9/CXCL10 signaling axis modulates the proliferation, migration and survival of tumor and endothelial cells. Nevertheless, the pro- or anti-tumoral effects depend on the pre-existing disease and the tumor entity [4]. Here, we provide in vitro and in vivo evidence that the absence of the CXCL10–CXCR3 signaling cascade reduces the proliferation rate of endothelial as well as tumor cells. This points to the overall tumor promoting effect of CXCL10 in the respective cell types during HCC progression. Since these effects are not dominant in vivo, there must be another player supporting these *Cxcl10*-dependent effects on tumor progression. Activated hepatic stellate cells (HSC) are the primary producer of ECM proteins. During liver injury and inflammation, they promote fibrogenesis, infiltrate the HCC stroma and contribute to HCC development. The enhancement of αSMA+ cells at the tumor site of the DEN/CCl_4_-treated *Cxcl10*^−/−^ mice suggests that these cells might be gaining a cancer-associated fibroblast-like (CAFs) function. It is known that CAFs are a heterogeneous cell population and secrete a vast repertoire of growth factors, cytokines and ECM components. Therefore, CAFs affect the behavior of the surrounding cells and remodel the ECM [22]. The enhanced amount and activation of these cells may be associated with the anti-tumoral microenvironment in the DEN/CCl_4_-treated *Cxcl10*^−/−^ mice compared to their WT counterparts [24].

Regarding the tumor-associated immune response, the DEN/CCl_4_-treated *Cxcl10*^−/−^ mice display a massive infiltration of anti-tumoral immune cells such as CD8+ T cells and NK cells in comparison to their WT counterparts. Of note, deletion of *Cxcl10* leads to a reduced number of apoptotic small mononuclear infiltrating cells in the area of the portal field in the DEN/CCl_4_-treated *Cxcl10*^−/−^ mice in comparison to the DEN/CCl_4_-treated WT mice. This is in line with previous studies showing that that CXCL10 induces apoptosis in T lymphocytes [23].

Furthermore, the enhancement of anti-tumoral immune cells in tumor-bearing *Cxcl10*^−/−^ mice induce a pro-inflammatory and therefore anti-tumorigenic microenvironment compared to corresponding tumor-bearing WT mice. It is known that chemokine receptor CXCR3 and its ligands have an impact on the inflammatory tumor microenvironment by recruiting different immune cell subpopulations [4]. Hirano et al. analyzed tumor-infiltrating lymphocytes in 44 HCC patients and found that the expression levels of CXCL9 and CXCL10 correlated with the number of infiltrating lymphocytes. Consequently, CXCL9 and CXCL10 promote lymphocyte infiltration into HCC and thus influence cancer immunology [26]. Consequently, there are also contradictory data. The anti-tumorigenic effects of CXCL9 are associated with leukocyte attraction to the TME in a paracrine fashion [4]. Therefore, individual parameters can alter the overall tumor microenvironment.

## 4. Materials and Methods

### 4.1. Cell Lines and Culture

To minimize culture-induced phenotypic changes, all cell lines were used within 2 weeks of thawing, and the same frozen batch was used for all experiments when possible. All cell lines were regularly tested for mycoplasma infection. Human HUH7 cells (well differentiated hepatocyte-derived carcinoma cell line) were cultured in DMEM supplemented with heat-inactivated 10% FCS and penicillin-streptomycin (100 U/mL). HUVECs (human umbilical vein endothelial cells) were cultured in Prigrow I supplemented with heat-inactivated 5% FCS and penicillin-streptomycin (100 U/mL). Both cell lines were cultured at 37 °C in a humidified atmosphere with 5% CO_2_.

### 4.2. Mouse Strains and Animal Model

Cxcl10^−/−^ mice were originally obtained from Jackson Laboratory, Bar Harbor, ME, USA, strain reference number RRID:IMSR_JAX:006087. In brief, for generation of these mice, a targeting vector containing a mouse phosphoglycerate kinase promoter driven neomycin resistance gene was used to replace exons 1–3 of the endogenous gene. The construct was electroporated into the 129S4/SvJae-derived J1 embryonic stem (ES) cells. Correctly targeted ES cells were injected into C57BL/6 blastocysts. The resulting chimeric mice were bred to C57BL/6 to generate mice heterozygous for the mutant allele and thereafter maintained on a C57BL/6J background in homozygous breedings by the vendor. For further details, see reference [27]. Upon purchase, breedings were established and maintained at the animal facility of the University of Aachen. C57BL/6J mice, also obtained from Jackson Laboratory, Bar Harbor, ME, USA, RRID:IMSR_JAX:000664, were used as wild type control mice. To induce fibrosis-associated hepatocarcinogenesis, mice of both genotypes were treated with a combination of mutagenic diethylnitrosamine (DEN) and the hepatotoxic carbon tetrachloride (CCl_4_). Mice received a single injection of DEN (100 mg/kg i.p.) at day 15 of age followed by repetitive injections of low-dose CCl_4_ (0.5 mL/kg i.p.) once a week (week 4–26). Untreated mice served as healthy control; CCl_4_-treated mice served as fibrotic controls. Mice were sacrificed 48 h after the last CCl_4_ injection at 26 weeks (wk) of age, and tumor burden (size and numbers/liver) was assessed.

Treatments were in accordance with the criteria of the German administrative panel on laboratory animal care. Animal studies were carried out in accordance with the law of the regional authorities for nature, environment, and consumer protection of North Rhine-Westphalia (Landesamt für Natur, Umwelt und Verbraucherschutz NRW (LANUV), Recklinghausen, Germany).

### 4.3. Hepatic Immune Cell Isolation and Flow Cytometry Analysis

Single-cell suspensions from WT and *Cxcl10*^−/−^ mice were isolated from freshly harvested murine livers by mechanical and enzymatic digestion as previously described [28]. Immune cells were enriched by density gradient centrifugation with 40/70 Percoll. Additionally, single-cell suspensions of fresh blood were collected via cardiocentesis. Hepatic as well as blood single cell suspensions were incubated with RBC lysis buffer for 1 min at room temperature. For flow cytometry analysis of the intrahepatic immune cell composition, single cell suspensions were stained with fluorochrome-conjugated antibodies for CD45, Ly6G, CD103, Ly6C, B220, NK1.1, CD11b, F4/80, CD11c, MHCII, CD3, CD4, CD8, CD25, CD62L, FoxP3 and CD115. For intracellular FoxP3 staining, cells were fixed in fixation/permeabilization buffer; 7AAD served as cells death marker for unfixed cells, the Fixable Viability Dye eFluor450 for fixed cells. The analysis of immune cell subpopulations (quantitative /absolute cell number and qualitative/percentage of CD45+ cells) were analyzed using an LSR Fortessa flow cytometery system. Absolute cell numbers were quantified via counting beads. Data were analyzed using FlowJo software.

### 4.4. H&E Staining

For H and E staining, formalin-fixed paraffin-embedded liver samples were cut into 5 μm thick slices. The samples were deparaffinized and rehydrated through xylol/descending alcohol series (100, 95, 70% ethanol and ddH_2_O). The sections were then stained with Mayer’s haemalum solution for 7 min at room temperature and afterwards rinsed in running tap water for several minutes. Subsequently, slides were counterstained with Eosin solution (1% (*w*/*v*)) for 5 min and dipped briefly in tap water to remove excess eosin. To dehydrate the stained sections, slides were immersed in ascending concentrations of ethanol (100, 95, 70% ethanol) and finally dipped three times in xylol. A mounting medium consisting of synthetic resins was used to seal the sections between slide and coverslip. Samples were then examined on a bright field microscope.

### 4.5. Immunofluorescence and Immunohistochemically Staining

Liver tissue were fixed in 4% neutral buffered formalin for 24–48 h, dehydrated in ethanol, and embedded in paraffin wax. Tissues were sectioned at 5 μm and transferred to positive-charged microscope slides. For the immunohistochemistry analysis, 5 μm tissue sections were deparaffinized and rehydrated through xylol/descending alcohol series (100, 95, 70% ethanol and ddH_2_O). Endogenous peroxidase activity was blocked methanol with 0.3% hydrogen peroxide for 10 min at room temperature. Heat-induced antigen retrieval was performed with citrate buffer (pH 6.0) in a steamer for 20 min. To block nonspecific staining, Avidin/Biotin Blocking Kit (Vector Laboratories, Burlingame, USA) was used; sections were stained for 1 h at room temperature with CXCL10, Ki-67, CD3 and cleaved caspase-3 antibodies, diluted in 1% BSA/PBS. Visualization of primary antibodies was performed via biotin-labeled secondary antibody and Peroxidase Substrate Kit DAB. Slides were counterstained with hematoxylin. CXCL10, Ki-67, CD3 and cleaved caspase-3 positive cells were counted (positive cells/total cells, three independent magnification fields per tumor/slide). Liver fibrosis was histologically assessed by quantification of Sirius red-positive non-tumor area per slide using the software Image J/NIH.

For the immunofluorescence staining analysis, liver tissue was fixed and frozen in Tissue-Tek O.C.T Compound. Tissues were sectioned at 5 μm and transferred to positive-charged microscope slides; 5 μm frozen liver tissue sections were fixed with 4% PFA/PBS and blocked with 5% BSA, 0,3% Triton X in PBS or PBS/2% azid for 1 h at room temperature. Sections were stained with CD4, CD8 and VEGF-R2/KDR overnight at 4 °C followed by a suitable secondary antibody. Antibodies were diluted in 1% mouse serum (PBS or PBS/2% azid). Cell nuclei were counterstained with DAPI. The percentage of the stained area, microvessels or cells were quantified in three independent magnification fields per tumor via using the software Image J/NIH.

### 4.6. Multiplex Staining

Stainings were performed on 4 µm-thick HCC tissue sections from the DEN/CCl4-treated WT and *Cxcl10*^−/−^ formalin-fixed and paraffin-embedded livers (IZFK UKA). For dewaxing, slides were heated for 1 h at 60 °C. Afterwards, sections were placed in 100% xylol followed by a rehydration procedure, with a descending ethanol series ending in dH_2_0. For fixation, sections were incubated for 20 min in 10% neutral buffered formalin solution (Sigma Aldrich, Darmstadt, Deutschland). After two additional washing steps, the slides were rinsed with 1X AR6 buffer (Perkin Elmer, Baesweiler, Germany) and heated to the boiling point using a microwave for antigen retrieval. Slides were washed afterwards two times in dH_2_O. The tissue sections were incubated for 30 min with Antibody Diluent/Block (Perkin Elmer, Germany). After the sections were prepared in the described fashion, the sequential antibody incubation was performed. In Table 1, the multiplex staining combinations are displayed according to their applied staining order.

The first primary antibody, diluted in Antibody Diluent/Block (manufacturer?), was added, and slides were incubated in the refrigerator at 4 °C overnight. Next, slides were washed three times with PBS Buffer before the secondary antibody incubation (Anti-rabbit HRP, 1:1500) for 45 min at room temperature. Afterwards, the slides were washed three times with TNT Buffer. The assigned TSA Plus dye (1:50, Akoya Bioscience, Marlborough, MA, USA) was added for 10 min. After another washing step with TNT-buffer, followed by two washing steps in dH_2_O, the slides were again heated in a microwave and the abovementioned staining was repeated for the next antigen until all desired antigens were stained with the assigned TSA Plus dyes. DAPI staining (1:500, Merck, Darmstadt, Germany) was performed by incubating the slides for 15 min at room temperature, followed by three additional washing steps in PBS. Then, the slides were embedded with Mowiol^®^ 4–88 (Carl Roth, Karlsruhe, Germany) and covered with coverslips.

### 4.7. Vectra Image Acquisition and in Form Analysis

Whole slide scans of the stained tissue sections were performed using the Vectra 3.0 Automated Quantitative Pathology Imaging System (PerkinElmer/Akoya Bioscience). For further analysis, regions of interest were annotated and identified in Phenochart (2046 × 1530 µm). Afterwards, the Vectra microscope acquired multispectral images in 40× magnification of these regions to achieve high resolution and to enable spectral unmixing for identification of up to 6 stains.

Using the inForm^®^ image analysis software, an algorithm for each staining was trained to phenotype and localize cells in the tumor microenvironment. To assign the cells to different tissue regions, the algorithm was trained to differentiate between tumor tissue (red), adjacent tissue (green), invasive margin (yellow) and blood vessels (blue) (Figure 5A). Hematoxylin and eosin stainings were performed to assure that tumor regions were assigned correctly. The invasive margin marks the region where immune cells enter the liver tissue from the blood vessels.

Based on the DAPI staining, cells were segmented and split into individual signals to differentiate between individual cells. We differentiated between CD4+ cells (red) and CD8+ cells (cyan), as displayed in Appendix A. During the phenotyping process, the inForm algorithm was trained by examples to differentiate between these cell types.

In addition, scoring analysis was performed. Scoring is based on a set threshold for the fluorescence intensity, which has to be passed, for a cell to be assigned positive. The scoring analysis also allows a differentiation between the different tissue segments, Appendix A. Scoring was performed for CD4+, CD8+, Ki67+ and cleaved caspase 3+ cells.

While the inForm analysis was carried out by using the “PhenoptR Addin” in R studios, quantification of αSMA and collagen IV expression was determined via area fraction using FiJi.

### 4.8. Proliferation Assay—BrDU Assay

The proliferation rate of human HUH7 cells and HUVEC after stimulation was determined using a Cell Proliferation ELISA, BrdU (colorimetric) following the manufacturer’s instructions. Cells were stimulated for 24 h with 0%, 0.5% and 10% heat-inactivated FCS (control) and 100 ng/mL CXCL9, 100 ng/mL CXCL10 and 100 ng/mL CXCL9 or 100 ng/mL CXCL10 with a CXCR3 Inhibitor/(±)-AMG 487 (End concentration 36 nM). All stimulants were diluted in DMEM or Prigrow supplemented with heat-inactivated FCS, penicillin-streptomycin (100 U/mL). The CXCR3 Inhibitor was pre-incubated for 1 h befor CXCL9 or CXCL10 stimulation.

### 4.9. cDNA Synthesis and Measurement of Gene Expression via qRT-PCR

Isolation of total RNA from snap-frozen liver tissue samples (tumorigenic livers were separated in tumor and surrounding tissue), in vitro-cultured bone marrow derived cells or hepatic endothelial cells was performed as previously described [28]. Quantitative real-time PCR reactions were performed with TaqMan Gene Expression Assays. Data were normalized to 18S expression and were analyzed by the −ΔΔCt method relative to control gene-of-interest expression (Primer list Appendix A).

### 4.10. Statistical Analysis

No statistical methods were used to predetermine sample size. Data are represented as means ± SEM. Continuous variables were compared by a two-sided T-test with Welch’s correction in case of unequal variances or one-way analysis of variance (ANOVA) with TukeyHSD or Bonferroni post hoc test. Values of *p* < 0.05 were considered significant. Statistical analyzes were performed with GraphPad Prism 9.0 (GraphPad, San Diego, CA, USA).

For details on Appendix A, please see Key resources table (Appendix A).

## 5. Conclusions

The chemokine CXCL10 shapes the intrahepatic tumor microenvironment, especially impacting the tumor-associated immune-responses during hepatic carcinogenesis. Genetic deletion of *Cxcl10* favors the infiltration of cytotoxic immune cells in the invasive margin and induces an anti-tumorogenic microenvironment. In addition, deletion of *Cxcl10* alters the chemokine/chemokine receptor network. Furthermore, *Cxcl10*^−/−^ impacts the tumor stoma composition, tumor cell proliferation and angiogenesis (Figure 8). Targeting the tumor microenvironment via CXCL10 is a potential novel therapeutic option for improving the efficiency of cancer therapies in HCC.

## Figures and Tables

**Figure 1 ijms-23-08112-f001:**
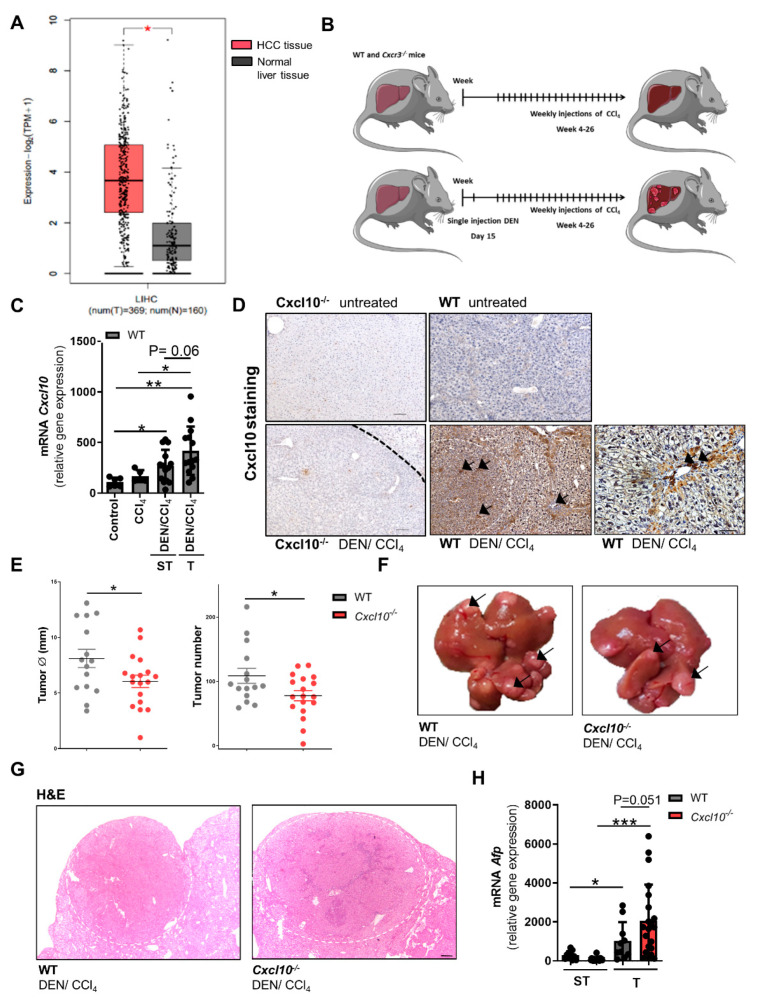
Intrahepatic expression of CXCL10 is elevated in HCC patients and contributes to hepatocarcinogenesis in mice: (**A**) intrahepatic *CXCL10* expression in human HCC patient’s gene expression profiling interactive analysis (GEPIA). HCC patients (red), normal liver tissue samples patients (grey)l (**B**) HCC model scheme: *Cxcl10*^−/−^ and WT litter mates were treated with a combination of DEN and the CCl4. Mice received a single injection of DEN (100 mg/kg i.p.) at day 15 of age followed by repetitive injections of low-dose CCl4 (0.5 mL/kg i.p.) once a week (week 4–26). CCl4-treated mice served as fibrotic control. Mice were sacrificed 48 h after the last CCl4 injection/after 26 weeks (wk) of age; (**C**) intrahepatic expression of *Cxcl10* mRNA in liver tissue of WT mice was measured by qRT-PCR (n ≥ 5). Tumor tissue (T) and surrounding tumor-free surrounding liver tissue (ST) were macroscopically separated. The amount of mRNA expression was normalized to the expression of *18S*; (**D**) exemplary pictures of immunohistological staining for/against CXCL10 in liver tissue of untreated as well as tumor-bearing WT and *Cxcl10*^−/−^ mice (H and E staining, scale bar 200 μm in far right lower picture, 100 μm for all other pictures, white/black dotted line marks margin between tumoral and surrounding tissue); (**E**) tumor burden of the DEN/CCl4-treated *Cxcl10*^−/−^ (n = 18) and WT mice (n = 16) were analyzed by counting and sizing [diameter in mm] of macroscopic HCC nodules; (**F**) representative pictures of livers from *Cxcl10*^−/−^ and WT mice. Black arrows: exemplary HCCs; (**G**) H and E staining of tumor-bearing liver tissue of the DEN/CCl_4_-treated WT and *Cxcl10*^−/−^ mice, size of tumors is not representative of overall tumor burden in the genotypes; scale bar 100 μm, same magnification in both pictures (**H**) mRNA expression level of *Afp* was quantified by qRT-PCR of non-tumoral liver tissue as well as tumor tissue (DEN/CCl_4_) of 26 wk old WT and *Cxcl10*^−/−^ mice. The amount of mRNA expression was normalised to the expression of *18S*. Data are expressed as mean ± SEM. Asterisks indicate statistical significance: * *p* < 0.05, ** *p* < 0.01, *** *p* < 0.001.

**Figure 2 ijms-23-08112-f002:**
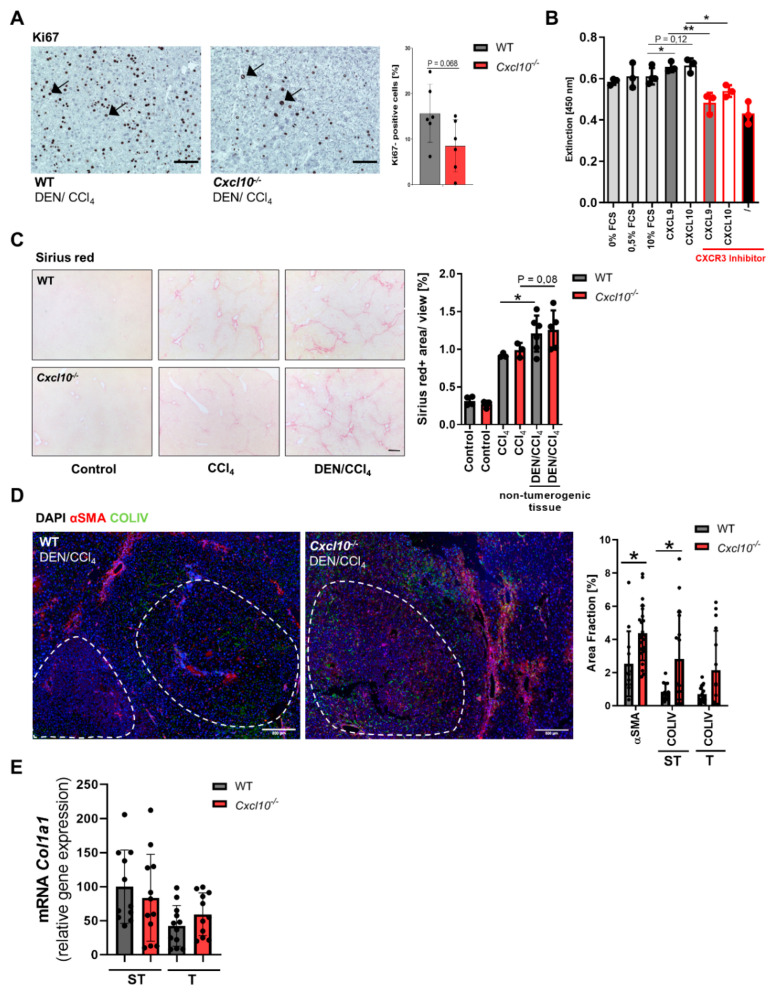
Genetic deletion of *Cxcl10* is associated with a reduced tumor cell proliferation and contributed to a rearrangement of the tumorigenic extracellular matrix composition: (**A**) Ki-67 staining of tumor-bearing livers from WT and *Cxcl10*^−/−^ mice; exemplary pictures of Ki-67-stained liver tissue of both genotypes (scale bar 200 μm); arrows: Ki-67+ tumor cells. The percentage of Ki-67+ cells/view field was determined. Three view fields were counted per sample; (**B**) human hepatoma cell line (HUH7) (n = 3) was stimulated with CXCL9 or CXCL10. Additionally, a CXCR3 inhibitor was added. Positive control: 0%, 0.5% and 10% FCS. After stimulation, a BrdU assay was performed, absorbance at 450 nm = proportional to proliferation rate; (**C**) representative Sirius red stainings of 26 wk old WT and *Cxcl10*^–/–^ mice (scale bar 100 μm) and quantification of Sirius red stainings (Sirius red+ area/view (%), (n ≥ 4); (**D**) multiplex imaging of tumorigenic WT and *Cxcl10*^–/–^ liver tissue, visualization of αSMA (red) and collagen IV (green), cell nuclei (blue). Area fraction of αSMA and collagen IV was calculated via Fiji; scale bar 500 μm (**E**) mRNA expression level Col1α1 was quantified by qRT-PCR of non-tumor liver surrounding tissue (ST) as well as tumor tissue (T) of WT and *Cxcl10*^−/−^ mice. The amount of mRNA expression was normalized to the expression of *18S*. Data are expressed as mean ± SEM. Asterisks indicate statistical significance: * *p* < 0.05, ** *p* < 0.01.

**Figure 3 ijms-23-08112-f003:**
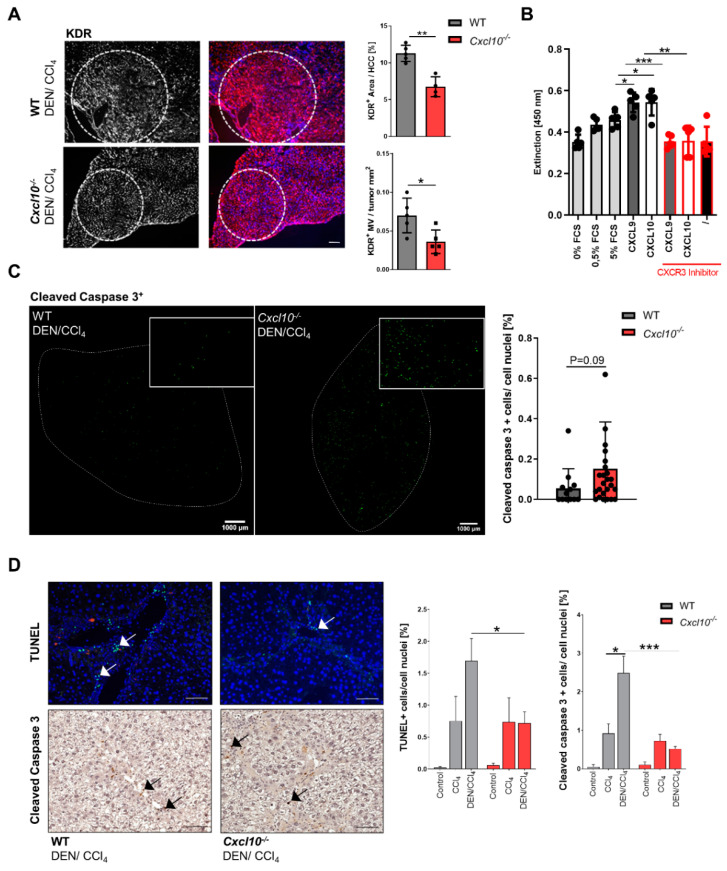
CXCL10 has a direct impact on the tumor-associated angiogenesis and alters the survival of small mononuclear infiltrating cells: (**A**) representative KDR/VEGF-R2 staining (red) of 26 wk old DEN/CCl_4_-treated WT and *Cxcl10*^−/−^ mice (scale bar 100 μm); nuclei were stained with DAPI (blue); white circle: tumor; area fraction and microvessel density (microvessel/tumor); (**B**) human hepatic endothelial cells (HUVEC) (n = 5) were stimulated with CXCL9 or CXCL10. Additionally, a CXCR3 inhibitor was added. Positive control: 0%, 0.5% and 5% FCS. After stimulation, a BrdU assay was performed. Absorbance at 450 nm = proportional to the proliferation rate; (**C**) cleaved caspase-3 staining of 26 wk old mice, activated caspase-3+ cells (green) per total cell count of tumor cells [%]; representative liver pictures; white cycles: tumor tissue; (**D**) TUNEL assay and cleaved caspase-3 staining of 26 wk old mice, TUNEL+/activated caspase-3+ cells per total cell count of infiltrating, mononuclear cells [%]. Three view fields were counted per sample; representative liver pictures (scale bar 200 μm); arrows: TUNEL+ or cleaved caspase-3+ cells. Data are expressed as mean ± SEM. Asterisks indicate statistical significance: * *p* < 0.05, ** *p* < 0.01, *** *p* < 0.001.

**Figure 4 ijms-23-08112-f004:**
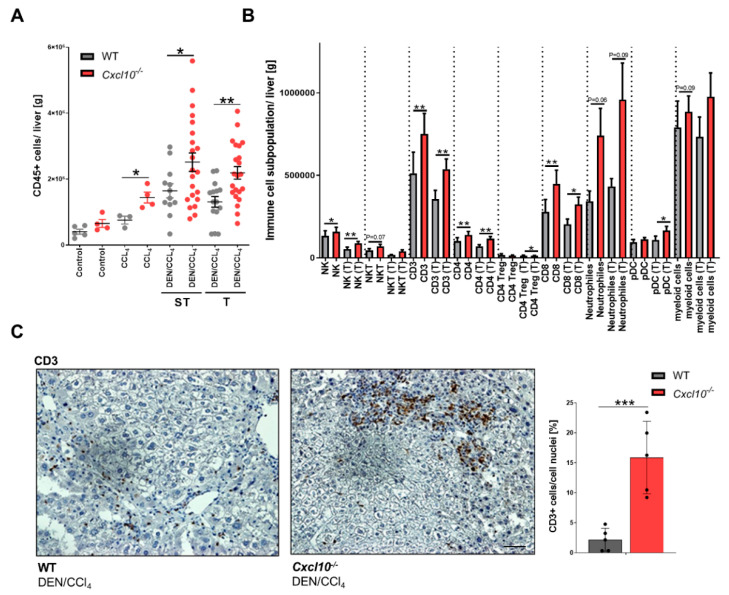
CXCL10 modulated the infiltration of immune cells to the tumor site: (**A**) quantification of intrahepatic CD45+ cells (CD45+ cells/gram liver tissue) was performed using multicolor flow cytometry analysis (n ≥ 3); (**B**) quantification of several intrahepatic immune cell subpopulations via multicolor flow cytometry analysis (immune cell subpopulation/gram liver); (**C**) quantification of CD3+ cells/nuclei [%] of the DEN/CCl4-treated WT animals and the *Cxcl10*^−/−^ animals (n = 5) via immunohistological staining. Three view fields were counted per sample. Scale bar 200 μm, same magnification in both settings. Data are expressed as mean ± SEM. Asterisks indicate statistical significance: * *p* < 0.05, ** *p* < 0.01, *** *p* < 0.001.

**Figure 5 ijms-23-08112-f005:**
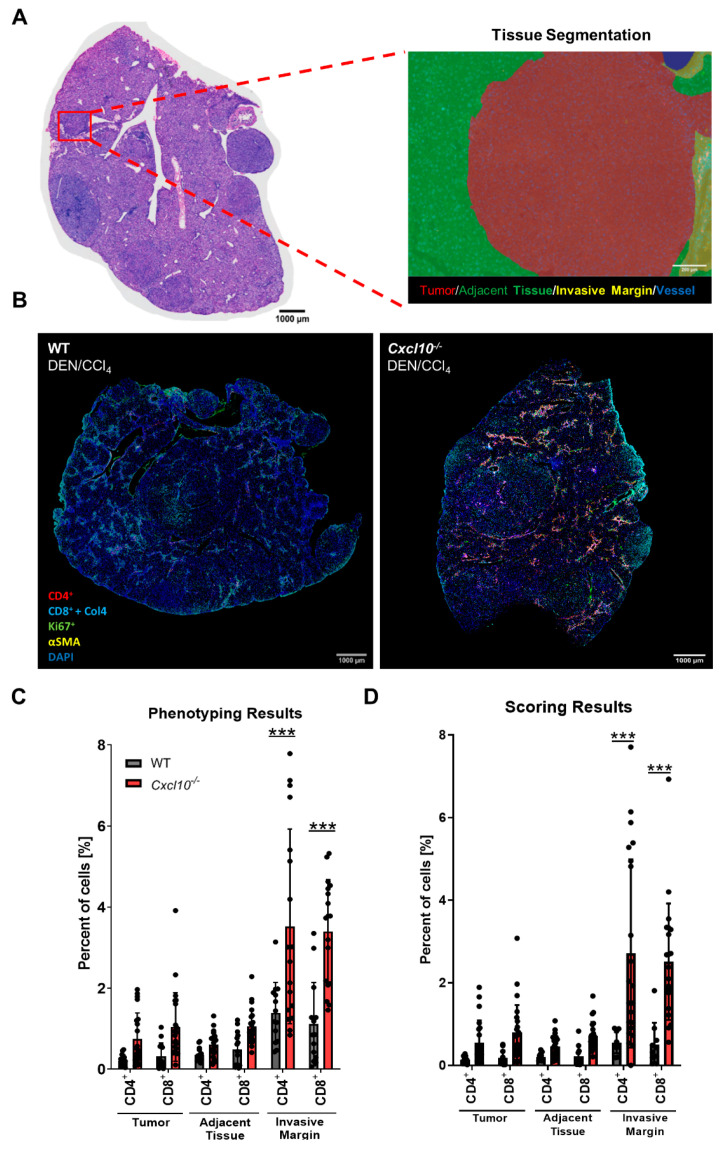
Deletion of *Cxcl10* promotes an accumulation of T cells in the HCC invasive margin: (**A**) representative H and E staining and tissue segmentation image of tumorigenic liver tissue. Tissue segmentation into tumor tissue (red), adjacent tissue (green), invasive margin (yellow) and blood vessel (blue); (**B**) multiplex staining of tumorigenic whole slide liver scans with Ki67 (green), CD4+ cells (red), CD8+ cells and collagen 4 (cyan), Ki67+ cells (green), αSMA (yellow) and DAPI (blue) of the DEN/CCl_4_-treated WT and *Cxcl10*^−/−^ mice; (**C**,**D**) quantification of CD4+ and CD8+ cells in liver tissue of the DEN/CCl_4_-treated WT and *Cxcl10*^−/−^ mice, phenotyping (**C**) and scoring (%, (**D**)) in different tissue regions of representative multiplex stained tissue slides using Fiji Software [21]. Data are expressed as mean ± SEM. Asterisks indicate statistical significance: *** *p* < 0.001.

**Figure 6 ijms-23-08112-f006:**
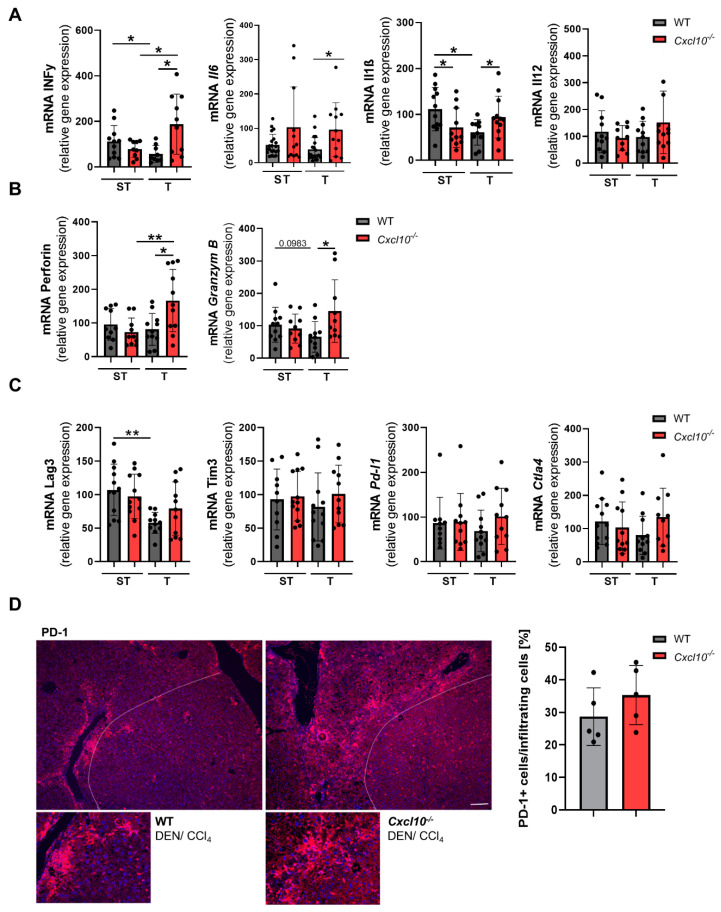
Altered inflammatory tumor microenvironment in *Cxcl10*^−/−^ mice in an experimental fibrosis-triggered HCC model; (**A**–**C**) to analyze the inflammatory microenvironment mRNA expression level of *Infy*, *Il6*, *Il1ß*, *Il12*, *Perforin*, *Granzym B*, *Pd-l1*, *Ctla-4*, *Lag3*
*and Timp3* was quantified by qRT-PCR in macroscopically separated fibrotic, non-tumor liver surrounding tissue (ST) as well as tumor tissue (T) of the DEN/CCl4-treated WT and *Cxcl10*^−/−^ mice. The mRNA expression was normalized to the expression of *18S*; (**D**) immunofluorescence staining and quantification of PD-1 of the DEN/CCl4-treated WT animals and *Cxcl10*^−/−^ animals (n = 5), (PD-1+ cells/DAPI-positive nuclei of infiltrating cells [%]), PD-1 (red), cell nuclei (blue) (scale bar 100 µm). Three view fields were counted per sample. Data are expressed as mean ± SEM. Asterisks indicate statistical significance: * *p* < 0.05, ** *p* < 0.01.

**Figure 7 ijms-23-08112-f007:**
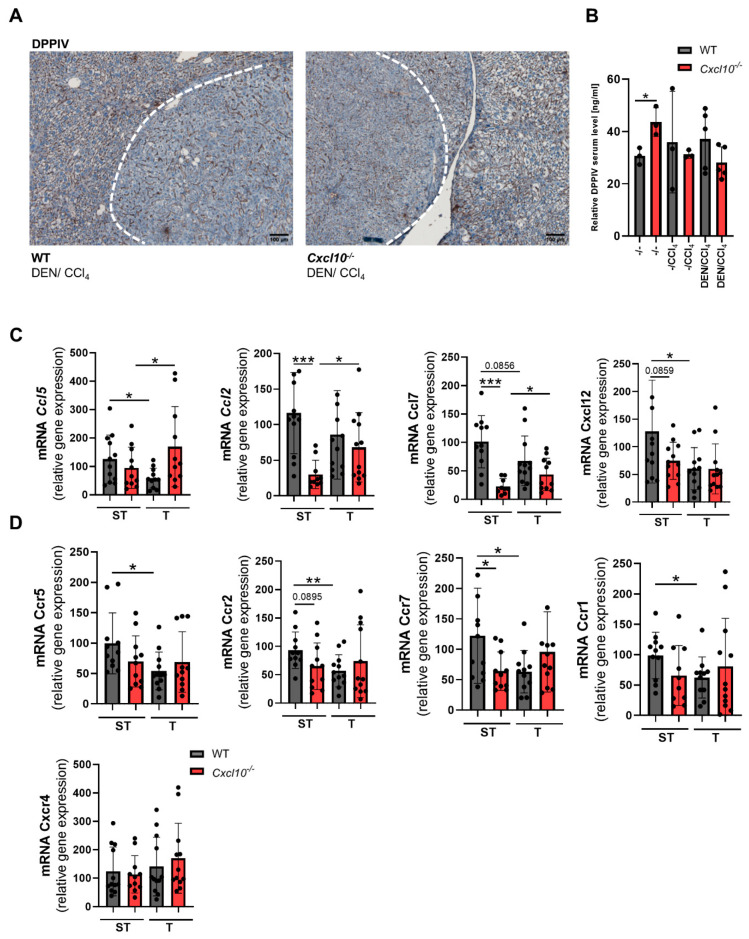
Genetic deletion of *Cxcl10* contributes to a rearrangement of the chemokine/chemokine receptor network: (**A**) immunohistological staining of DPPIV of the DEN/CCl4-treated WT animals and *Cxcl10*^−/−^ animals (scale bar 100 µm); (**B**) quantification of serum DPPIV via ELISA of all experimental groups (n ≥ 3); (**C**,**D**) characterization of the inflammatory chemokine/chemokine receptor network via qRT-PCR; mRNA expression level of *Ccl5*, *Ccl2*, *Ccl7 Ccr5*, *Ccr2*, *Ccr7*, *Ccr1* and *Cxcr4*. The mRNA expression was normalized to the expression of *18S*. Data are expressed as mean ± SEM. Asterisks indicate statistical significance: * *p* < 0.05, ** *p* < 0.01, *** *p* < 0.001.

**Figure 8 ijms-23-08112-f008:**
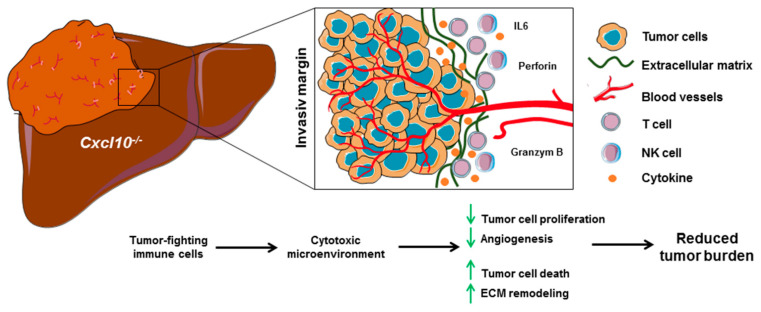
The chemokine CXCL10 shapes the intrahepatic tumor microenvironment during fibrosis-associated HCC progression. Deletion of *Cxcl10* promotes the infiltration of tumor-fighting immune cells to the tumor site. Particularly, T cells and NK cells form a destructive boundary around the HCCs and induce a cytotoxic microenvironment. Furthermore, CXCL10 affects the entire chemokine/chemokine receptor network, tumor stoma composition, tumor cell proliferation and angiogenesis. Taken together, these effects lead to a reduced tumor burden. CXCL10 is a promising therapeutic target for the modulation of the tumor microenvironment.

**Table 1 ijms-23-08112-t001:** Antibody combinations and order for multiplex staining.

	Antibody	Manufacturer	TSA Dye	Manufacturer
Cleaved Caspase 3	CD4	Abcam	Cy 3.5	Perkin Elmer/Akoya Bioscience
	CD8	Bioss	Cy 5	
	Cleaved Caspase 3	Abcam	FITC	
Ki67	aSMA	Abcam	Cy 3	Perkin Elmer/Akoya Bioscience
	CD4	Abcam	Cy 3.5	
	Col4	Novotec	Cy 5.5	
	CD8	Bioss	Cy 5	
	Ki67	Abcam	FITC	

## Data Availability

All source data of this study can be obtained from the corresponding author on reasonable request.

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
