# Peer review of "Chemokine CXCL10 Modulates the Tumor Microenvironment of Fibrosis-Associated Hepatocellular Carcinoma"

_ijms, 2022, doi:10.3390/ijms23158112_

Round 1

Reviewer 1 Report

In the study entitled "Chemokine CXCL10 modulates the tumor microenvironment of fibrosis-associated Hepatocellular carcinoma" by Brandt et al., the authors demonstrated that CXCL10/CXCR3 axis is involved in the development of HCC by reshaping the tumor microenvironment and especially modulating the recruitment of anti-tumor immune cells. This is an overall interesting and well-written study that will definitely be of interest for people in the field and that coud potentially pave the way for new treatments against HCC. I have nonetheless some concerns that might be addressed by the authors.

1) The authors used a Cxcl10 KO mouse model to test the involvment of CXCL10 in the development of HCC but do not appropriately describe how the KO was made. This has to be done at least in the methods section. Besides, an IHC against CXCL10 would be appreciable to show that these mice do not express CXCL10.

2) The authors used HCC RNA-seq data to show that CXCL10 expression is increased in HCC patients. HCC is heterogeneous in terms of etiology. Is this increase observed whatever the causative agents?

3) The authors observed an increased of intrahepatic Cxcl10 mRNA level after treatment of mice with fibrosis induced agents. The liver is composed of different cell types. Is this increase observed in all cell types? Is it confirmed at the protein level? This information is not very clear from the figure 1d as only IHC for WT mice treated with DEN and CCL4 is displayed. It would be important to display the images for all the conditions.

4) The authors propose that the small effect of CXC10 KO on tumor cell proliferation could be due to compensatory effects by CXCL9, another ligand of CXCR3. Is CXCL9 expression also increased in HCC and in DEN/CCL4 treated mice? Is CXCL9/CXCL10 double repression triggers additional effect on hepatoma cell lines compared to single repression?

Minor comments

- It would be appreciated if the authors could include the legend directly on the figures especially for figure 1 with HCC patients and healthy individuals.

- lane 64: be is missing between appears to and dependent  

Reviewer 2 Report

It has been well established that CXCL10 participates in crosstalk between hepatocytes and liver fibrosis. Therefore, it cannot be argued that the emphasis on the importance of CXCL10 in fibrosis-associated hepatocellular carcinoma is a completely new field. Moreover, in figures 1G and H, the tumor volume occupying the liver was rather increased in CXCL10-/-, and AFP level was also significantly increased. In Figure 2C, there is no difference in the fibrosis tendency in the liver tissue between the groups. Rather, in figure 2D, the results for increased expression of a-SMA and collagen IV in the CXCL10-/- group is also awkward. CXCL10 is a pro-fibrotic factor, so if it is reduced, the expression of a-SMA or collagen IV will be reduced due to the inability to construct liver fibrosis, followed by a decrease in AFP expression compared with CXCL10+/+.

Round 2

Reviewer 1 Report

The authors properly addressed my comments

Reviewer 2 Report

The issues of this reviewer's doubts have been cleared by the authors' responses.  There is no additional issue to raise.